# The Emerging Role of Cohesin in the DNA Damage Response

**DOI:** 10.3390/genes9120581

**Published:** 2018-11-28

**Authors:** Ireneusz Litwin, Ewa Pilarczyk, Robert Wysocki

**Affiliations:** Institute of Experimental Biology, University of Wroclaw, 50-328 Wroclaw, Poland; pilarczyk.e@gmail.com (E.P.); robert.wysocki@uwr.edu.pl (R.W.)

**Keywords:** cohesin, cohesin loader, DNA double-strand breaks, replication stress, DNA damage tolerance

## Abstract

Faithful transmission of genetic material is crucial for all organisms since changes in genetic information may result in genomic instability that causes developmental disorders and cancers. Thus, understanding the mechanisms that preserve genome integrity is of fundamental importance. Cohesin is a multiprotein complex whose canonical function is to hold sister chromatids together from S-phase until the onset of anaphase to ensure the equal division of chromosomes. However, recent research points to a crucial function of cohesin in the DNA damage response (DDR). In this review, we summarize recent advances in the understanding of cohesin function in DNA damage signaling and repair. First, we focus on cohesin architecture and molecular mechanisms that govern sister chromatid cohesion. Next, we briefly characterize the main DDR pathways. Finally, we describe mechanisms that determine cohesin accumulation at DNA damage sites and discuss possible roles of cohesin in DDR.

## 1. Introduction

Genomes of all living organisms are continuously challenged by endogenous and exogenous insults that threaten genome stability. It has been estimated that human cells suffer more than 70,000 DNA lesions per day, most of which are single-strand DNA breaks (SSBs) [1]. Moreover, in every human cell ≈50 DNA double-strand breaks (DSBs), considered to be the most cytotoxic DNA lesions, arise during each S-phase through passive conversion of SSBs to DSBs via replication machinery [2,3,4]. If left unrepaired, such damage may result in chromosome loss and/or cell death. On the other hand, inaccurate repair of DNA damage may lead to point mutations, loss of heterozygosity, or gross chromosomal rearrangements that can drive cancerogenesis [5,6]. To avoid such deleterious outcomes, all eukaryotic cells have evolved DNA damage response (DDR) mechanisms that sense and repair DNA damage or allow them to complete replication of damaged DNA. These include DNA damage checkpoint (DDC), DNA damage repair, and DNA damage tolerance pathways (DDT) [7,8,9].

Cohesin is a multiprotein, ring-shaped complex that was first characterized in budding yeast *Saccharomyces cerevisiae*. Its canonical role is to tether sister chromatids together until the onset of anaphase to prevent premature sister chromatid separation and ensure equal segregation of genetic material to daughter cells [10,11]. Moreover, cohesin is crucial for regulation of transcription and 3D chromatin organization [12,13,14]. However, even before the role of cohesin in sister chromatid cohesion or genome architecture was documented, research on fission yeast *Schizosaccharomyces pombe* revealed that disruption of cohesin results in increased sensitivity to UV and γ-irradiation [15]. Presently, cohesin is emerging as one of key factors involved in the response to DNA damage.

## 2. Molecular Architecture of the Cohesin Complex

In all eukaryotic organisms, the cohesin complex consists of three core, essential subunits: two structural maintenance of chromosome proteins (SMC), Smc1 and Smc3, and one non-SMC protein, Scc1 (RAD21 in humans). The SMC subunits are composed of an N-terminal part containing the Walker A motif, a coiled-coil region separated in the middle by a globular domain called the hinge, and a C-terminal part that includes the Walker B motif [16]. The N- and C-terminal halves fold back on themselves to form intramolecular, antiparallel coiled-coils and functional ATPase called the head domain. Smc1 and Smc3 interact with each other through the hinge and the head domains creating an oblong heterodimer. Scc1 bridges both Smc subunits by binding Smc3 through its N-terminal part and by interacting with Smc1 via its C-terminus (Figure 1). Together, Smc1, Smc3, and Scc1 create a ring-like structure that is about 50 nm long with a ≈35 nm diameter [17,18,19,20]. In addition, there are several other proteins that associate with the cohesin ring, including two essential HEAT proteins, Scc3 (SA1 and SA2 in humans) and Pds5 (PDS5A and PDS5B in humans), that interact with the C-terminal and the N-terminal part of Scc1, respectively, and Wpl1 (WAPL in humans), which is a non-essential, unstably bound protein that interacts with cohesin through Scc3, Pds5, Scc1, and Smc3 (Figure 1) [21,22,23,24].

## 3. Sister Chromatid Cohesion Process

### 3.1. Cohesin Loading

The first stage of the sister chromatid cohesion process is cohesin loading. In *S. cerevisiae*, cohesin binding to chromatin starts in late G1/early S-phase when Scc1 expression is reinduced [25,26]. In humans, on the other hand, intact cohesins, that were deposited by WAPL in prophase, associate with chromatin in telophase [27,28]. Both in yeast and humans, cohesin loading depends on the Scc2–Scc4 cohesin loading complex (NIPBL^A^- or NIPBL^B^-MAU2 in humans) that allows entrapment of sister chromatids inside the cohesin ring [20,26,29,30,31,32]. Scc2 is a large protein (171 kDa) that contains N- and C-terminal globular domains separated by HEAT repeats [33]. Scc4 is a smaller protein (72 kDa) that consists mostly of TPR (tetratricopeptide repeat) structural motifs [34]. It has been shown, both in humans and in yeast, that Scc4 binds to the unstructured N-terminal part of Scc2, creating a stable, heterodimeric hook-shaped complex (Figure 2) [34,35,36].

Biochemical analysis revealed that Scc2 is a DNA binding protein that interacts poorly with single-stranded DNA (ssDNA) but has a high affinity to double-stranded DNA (dsDNA) and Y-form DNA. Interestingly, it has been recently shown that the C-terminal part of Scc2 is sufficient to perform the cohesin loading reaction in vitro, but in vivo the presence of Scc4 is strictly required [32]. At the molecular level, cohesin loading requires physical interactions between the cohesin loader and cohesin. Topological binding of DNA also requires functional Smc1 and Smc3 ATPases as the inability to hydrolase ATP abolishes cohesin loading [32,37,38,39]. Entrapment of chromatids inside the cohesin ring requires temporary opening of the ring; however, localization of an entry gate is still under debate. Artificial tethering of Smc1 and Smc3 hinge domains has been shown to prevent cohesin association with chromatin. This result suggests that the transient opening of the hinge domains allows DNA molecules to enter the cohesin ring (Figure 2) [40]. Alternatively, the Smc3–Scc1 interface was proposed to work as an entry gate for DNA [17]. In both cases, the role of Scc2 would be at least to promote ATP hydrolysis to drive opening of the cohesin ring, possibly through considerable changes in cohesin conformation [38,40]. Interestingly, recent data suggest that sister chromatid entrapment within the cohesin ring may require two cycles of cohesin loading, both taking place behind replication forks: first on dsDNA generated by leading strand synthesis and the second on ssDNA temporarily present at the lagging strand [41].

Genome-wide binding maps of the Scc2–Scc4 complex revealed that the cohesin loader associates with multiple chromosomal locations, including chromosome arms, centromeres, and telomeres [42,43,44]. Because cohesin loading does not depend on DNA sequence, the Scc2–Scc4 complex seems to be attracted to specific loading sites through interaction with other proteins [32]. In humans, cohesin loader association with chromatin in early S-phase requires physical interaction between NIPBL-MAU2, the MCM helicase and the Dbf4-dependent kinase (DDK) [45]. In contrast, it seems that in yeast Scc2–Scc4 binds to chromatin independently of the pre-replication complex formation and only partially requires DDK, although cohesins are transiently present at active replication sites [46,47]. It has been recently shown that in budding yeast recruitment of the cohesin loader to centromeric regions depends on Scc4 interaction with the Ctf19 kinetochore protein phosphorylated by DDK. This allows efficient accumulation of cohesin at pericentromeric regions and establishment of centromeric cohesion [34,48,49]. Interestingly, mutations in components of the Ctf19 kinetochore complex reduce cohesin levels only at centromeres and are not lethal [34,49]. Moreover, Scc2–Scc4 interaction with chromatin also occurs in the G2 phase when most of the pre-replication (pre-RC) complex components are uncoupled from chromatin [50,51]. This suggests that other pathways, mediating cohesin loading complex recruitment to specific chromosomal locations, must exist. Such an additional mode of cohesin loader recruitment may depend on chromatin remodelers. Recently, Lopez-Serra et al. revealed that the budding yeast RSC (PBAF in humans) chromatin remodeling complex is required for Scc2–Scc4 association with chromatin [44]. RSC colocalizes with Scc2–Scc4 on chromatin and both complexes cooperate to create nucleosome-free regions that are important for cohesin loading. Conversely, lack of functional RSC disrupts the association of the cohesin loader with the chromatin, leading to decreased amounts of cohesin at centromeres and chromosome arms and severe loss of sister chromatid cohesion [44]. In addition, Irc5 (LSH or HELLS in humans), a putative chromatin remodeler, was also shown to be involved in cohesin loading in budding yeast. It was demonstrated that lack of Irc5 or its translocase activity results in disrupted interaction between Scc2, chromatin, and the cohesin complex. This leads to decreased levels of chromatin-bound cohesin at centromeres, chromosome arms, and the rDNA region, causing mild premature sister chromatid separation [52]. In humans, both the SNF2 subunit of chromatin remodelers and the ATRX chromatin remodeler promote cohesin association with chromatin, but whether their role is to attract the cohesin loader is currently unknown [53,54,55]. Just after loading onto chromatin, yeast and human cohesin complexes are pushed away by transcription machinery from loading sites to multiple chromosomal locations, which are often regions of convergent transcription. Cohesin translocation seems to be independent of loading cycles but depends instead on sliding along chromatin [43,56,57]. It is worth mentioning that cohesin is crucial for the global transcription regulation and interphase chromatin architecture. It seems that cohesin has the ability to create and extend DNA loops allowing long range chromatin interactions. In human cells, cohesin often colocalizes with the CTCF transcription factor, which regulates the extend of loop extrusion. Together, these two factors promote gene expression by establishing promoter–enhancer interactions and enable compartmentalization of the genome into topologically associating domains [12,13,14,58].

### 3.2. Cohesion Establishment and Maintenance

Once loaded, cohesin interaction with chromatin is not stable and can last only a few seconds [59,60,61]. The dynamic association of cohesin with chromatin depends at least on Wpl1, Pds5, and the acetylation state of Smc3. It was proposed that the DNA entrapped within the cohesin ring may interact with unacetylated Smc3 lysine residues, K112 and K113 (Lys105/Lys106 in humans), leading to ATP hydrolysis and diminished interaction between head domains [17]. Next, Wpl1 and Pds5 destabilize the Smc3–Scc1 interface by dissociating the N-terminal part of Scc1 from the Scc3 head, creating an exit gate for the DNA [17,62]. In budding yeast, stable sister chromatid entrapment takes place in S-phase and relies on acetylation of Lys112 and Lys113 located on the Smc3 head. This reaction was shown to be mediated by the Eco1 acetyltransferase, which interacts with the PCNA replisome component (Figure 2) [63,64,65]. In humans, two acetyltransferases, ESCO1 and ESCO2, acetylate Smc3, but recent data suggest that only ESCO2 is crucial for cohesion establishment under normal conditions [66]. Acetylation of Lys112/Lys113 most likely prevents ATP hydrolysis stimulated by the DNA trapped within the cohesin ring precluding WAPL from the Smc3–Scc1 interface opening [17,67,68]. Interestingly, cohesion establishment is strongly influenced by proteins that associate with replication forks, including Chl1, Csm3, Ctf4, Ctf18, Mrc1, Tof3, or the primase; lack of any of these factors results in significant precocious sister chromatid separation. The exact role of these proteins in cohesion establishment is largely unknown, but they might provide optimal replisome architecture for efficient Smc3 acetylation [69,70,71,72]. Later in the cell cycle of yeast, cohesion is sustained by Pds5 and Scc3. How these proteins maintain cohesion is not clear but Pds5 may protect Smc3 Lys112 and Lys113 from deacetylation and/or preclude polysumoylation of Scc1 and its proteosomal degradation (Figure 2) [73,74,75]. In humans, cohesion is maintained by Sororin, which is exclusively present in vertebrates. It was proposed that Sororin blocks WAPL binding to Pds5 preventing cohesin deposition from chromatin [76].

### 3.3. Cohesion Dissolution

To allow sister chromatid segregation between daughter cells, cohesin needs to be removed from chromatin. In budding yeast, after degradation of Pds1 securin (PTTG1 in humans), the Esp1 separase (ESPL1 in humans) is free to cleave Scc1, creating an exit gate for sister chromatids [25]. This allows Hos1 (HDAC8 in humans) to deacetylate Smc3, allowing fast and efficient genome-wide removal of cohesin from chromatin (Figure 2) [77,78]. In humans, the majority of the cohesin complexes are released from chromatin by WAPL in prophase and only a subset of cohesin, especially centromere-bound, is cut by the separase. The cohesin complexes removed by WAPL are not degraded but are loaded again on chromatin already in telophase [79,80].

## 4. DNA Damage Response Mechanisms

### 4.1. Signaling and Repair of DNA Double-Strand Breaks

DNA double-strand breaks are the most hazardous type of DNA damage because they compromise the physical integrity of chromosomes, threatening genomic stability. DNA double-strand breaks can be induced endogenously by reactive oxygen species, replication through ssDNA gaps or generated during physiological processes such as mating type switching in yeast or V(D)J recombination in mammals. This type of DNA damage can also be induced exogenously, mostly due to exposure to genotoxic chemicals or radiation [81]. In eukaryotes, two major pathways allow DSB repair: homologous recombination (HR) and nonhomologous end joining (NHEJ). In yeast, HR is a major mechanism of DSB repair, whereas NHEJ plays a minor role in this process. In mammals, on the other hand, NHEJ repairs most of the DSBs and HR is mainly used to repair replication fork-associated breaks [82,83]. Moreover, in response to DSBs, eukaryotic cells activate survival mechanisms called DNA damage checkpoints that regulate cell cycle progression, gene transcription, DNA repair, and apoptosis induction [7,9]. Importantly, defects in these processes lead to many diseases, including cancers, developmental disorders, and immune deficiencies.

#### 4.1.1. DNA Damage Checkpoint Activation in Response to DNA Double-Strand Breaks

DDC are complex mechanisms that enable to sense DNA, amplify, and transduce the signal of DSB presence. DDC operates at the G1/S boundary, in S-phase and at the G2/M transition point. Activation of DDC results in cell cycle arrest or replication slowing giving additional time for damage repair, regulates transcription of DNA repair factors, and prevents exhaustion of proteins involved in damage repair. If DNA damage cannot be repaired, DDC may induce cellular senescence or apoptosis [9,84].

DSBs are first recognized by the Mre11-Rad50-Xrs2 (MRX) complex (MRE11-RAD50-NBS1 or MRN in humans) that binds and holds ends of broken DNA together [85,86]. Moreover, MRX recruits the sensor kinase Tel1 (ATM in humans) through the C-terminal part of Xrs2 [87,88]. Tel1 belongs to the conserved phosphoinositide 3-kinase-related protein kinase (PIKK) family and phosphorylates target proteins on Ser or Thr residues followed by Glu (S/T-Q motif). Immediately after its recruitment, Tel1 phosphorylates histone H2A on Ser129 (Ser 139 in humans, H2A-P) in the vicinity of the DSB [89]. This allows the adaptor protein Rad9 (53BP1 in humans) to interact with H2A-P through BRCT domains and become phosphorylated by Tel1 [90,91]. Phosphorylated Rad9 is then bound by Rad53 (CHK2 in humans) and Chk1 (CHK1 in humans) effector kinases through FHA and CAD domains, respectively. Both proteins associated with Rad9 become phosphorylated by Tel1 [92,93,94]. Hyperphosphorylated Rad53 and Chk1 dissociate from Rad9 and phosphorylate their targets [95]. In humans, the Tel1 homolog ATM is also recruited to a damage site by the MRN complex, where it phosphorylates histone H2AX on Ser139 (γH2AX) [85,96,97]. This in turn attracts the Rad9 ortholog MDC1 to the damage sites and allows its phosphorylation by ATM. Importantly, MDC1 is crucial for stable accumulation of other adaptor proteins, e.g., 53BP1 that is important for full CHK2 activation [98,99,100,101].

In budding yeast, the majority of DSBs generated in the S or G2 phase are subjected to DNA resection, which is a process of enzymatic degradation of 5′ ends of broken DNA. DNA resection is initiated by MRX and further catalyzed by the Exo1 exonuclease or the Dna2 nuclease in a complex with the Sgs1 helicase (BLM or WRN in humans) [102,103]. Next, long 3′ ssDNA tails are immediately coated by the replication protein A (RPA) complex that prevents formation of secondary structures and unlicensed ssDNA degradation [104]. RPA-coated ssDNA overhangs generated during DNA resection disrupt Tel1/ATM-dependent signaling and allow Ddc2 (ATRIP in humans) protein to bind to RPA and recruit a second PIKK, Mec1 (ATR in humans). Since RPA-bound ssDNA spreads thousands of base pairs from DSB, many Mec1/ATR molecules associate with processed DNA, generating a strong checkpoint signal [85,105,106,107,108,109,110,111]. In response to DSB, ATR phosphorylates H2AX histone and activates effector kinases, mainly CHK1. However, it seems that replication stress signaling is the main role of this PIKK [112]. Activation of effector kinases leads to cell cycle-dependent phosphorylation of hundreds of proteins, leading to cell cycle arrest, transcription activation of repair factor genes, or even induction of apoptosis [84,113].

#### 4.1.2. Repair of DNA Double-Strand Breaks by Homologous Recombination in Mitotic Cells

Homologous recombination is initiated by DNA resection and formation of RPA-coated ssDNA. Because RPA has higher affinity to ssDNA than Rad51 recombinase, specialized proteins (e.g., Rad52 mediator) are needed to discard RPA [114,115]. The fully formed nucleofilament is a stretched, elongated structure, consisting of Rad51 molecules bound to ssDNA, which has the capacity to perform a whole genome search for homologous DNA [116,117]. Mitotic cells preferentially use sister chromatids as a donor template for HR because of very low probability of heterozygosity loss. Alternatively, repair can also be templated from a homologous chromosome but in the case of crossover, loss of heterozygosity often occurs. In some cases, ectopic sequences, often located on nonhomologous chromosomes, can be used as donors but often at the cost of gross chromosomal rearrangements [118,119]. Once a homology has been found, the nucleofilament displaces donor dsDNA and pairs with the complementary sequence, creating a displacement-loop (D-loop) structure. Next, the free 3′ DNA end of the filament primes DNA synthesis restoring lost chromosome sequence [120]. In yeast mitotic cells, most D-loops are disrupted shortly after initiation of DNA synthesis. Quick termination of D-loop progression precludes formation of recombination structures that may give rise to crossover products [121,122,123]. Failure to disrupt the D-loop early may lead to further D-loop extension and allow formation of a four-way branched DNA structure called a double-Holliday junction (dHJ) [124]. By the time of mitosis, all dHJs must be disbanded to allow separation of sister chromatids. Mitotic cells preferentially use the Sgs1-Top3-Rmi1 complex (BLM-TOP3α-RMI1-RMI2 complex in humans) that dissolves dHJs without crossing over [125]. Alternatively, dHJs can be cleaved by Mus81-Mms4 (MUS81-EME4 in humans), Slx1-Slx4 or Yen1 (GEN1 in humans) nucleases but often at the cost of crossover [126].

#### 4.1.3. Nonhomologous End Joining

As stated before, NHEJ is the main mechanism of DSB repair in humans. In contrast to HR, NHEJ is not restricted by the cell cycle phase because it does not require homologous DNA as a template for repair. Instead, NHEJ provides fast and efficient machinery that enables processing and ligation of broken DNA ends, albeit, often at the cost of small deletions or insertions.

The KU complex is a key NHEJ factor that consists of two subunits, Ku70 and Ku80. KU recognizes DSB and accommodates broken DNA ends and protects against DNA resection [127,128,129,130]. Moreover, its stable association with DNA allows other proteins to be recruited to the damage site including DNA end-processing factors. Next, DNA end cleaning enzymes (e.g., Artemis, KU, tyrosyl DNA phosphodiesterases, Pol λ, Pol η) process broken DNA ends to allow DNA ligation [131,132,133,134,135,136,137,138,139]. Importantly, processing of DNA termini sometimes requires addition or removal of several DNA bases, leading to small deletions or insertion, thus driving mutagenesis. In the last step, the ligase4-XRCC4 complex joins the broken DNA ends, reestablishing chromosome continuity. It should be noted, however, that in certain situations, NHEJ may occur in the absence of the KU complex in a process referred to as alternative end joining (a-EJ). Like canonical NHEJ (c-NHEJ), a-EJ is a process based on broken DNA ligation but in contrast to c-NHEJ requires limited DNA end resection. Importantly, it was shown that a-EJ leads to deleterious chromosomal rearrangements including translocations as well as intra- and interchromosomal deletions and insertions that are normally prevented by KU [140,141,142].

### 4.2. DNA Damage Tolerance Mechanisms

DNA replication is a vital process often challenged by various impediments that arise from both endogenous and exogenous sources. Stalling of replication fork progression can be potentially dangerous as prolonged exposure of ssDNA greatly increases the risk of DNA base damage and fork collapse [4,143]. To allow for the bypass of replication-blocking lesions and completion of DNA synthesis, cells employ sophisticated DNA damage tolerance mechanisms. In budding yeast, three DDT pathways operate: DNA translesion synthesis (TLS), template switch (TS), and HR. The first two are activated by proliferating cell nuclear antigen (PCNA) mono- and polyubiquitylation, while the third one is actively prevented at replication forks via PCNA sumoylation [8].

Under replication stress conditions, RPA-coated ssDNA regions generated at or behind stalled replication forks recruit the Rad18 ubiquitin ligase (E3), which together with the Rad6 ubiquitin-conjugating enzyme (E2), monoubiquitylates PCNA on Lys164. Next, monoubiquitylated PCNA attracts translesion polymerases that replicate across DNA lesion due to more specious active site that allows accommodation of variety of modified DNA bases. However, TLS polymerases lack proofreading activity, which leads to an increased frequency of misincorporated nucleotides and largely accounts for genome-wide mutagenesis [144,145,146,147]. Also, in humans, PCNA monoubiquitylation at Lys164 enables recruitment of TLS polymerases to damaged DNA sites. However, in contrast to yeast, this reaction can be mediated not only by the Rad6-Rad18 complex but also by other ubiquitin ligases, such as RNF8 or CRL4Cdt2 [148,149,150,151]. PCNA can be further polyubiquitylated through the addition of Lys63 linked ubiquitin chains to monoubiquitylated PCNA triggering a recombination-based TS mechanism. In budding yeast, PCNA polyubiquitylation is performed by Mms2-Ubc13 (E2) and Rad5 E3 enzyme while in humans at least two ubiquitin ligases, HLTF and SHPRH, promote TS. Template switch allows the undamaged sister chromatid to be used as a temporary template to resume and/or complete replication [146,152,153,154,155,156]. In budding yeast, TS seems to operate mainly uncoupled from the replication forks and is crucial for postreplicative ssDNA gap filling [154,157,158]. Moreover, TS requires presence of Rad52 and Rad51 proteins that enable formation of pseudo-dHJ structures (often called sister-chromatid junctions or X-shaped molecules) [154,159,160,161]. These structures are mainly dissolved by the Sgs1-Top3-Rmi1 complex during S-phase, while the remaining structures can be resolved by Mus81 or Slx4 endonucleases, but often at the cost of crossover [161,162,163].

In certain situations, replication can be resumed and completed by canonical HR, which requires Rad52 and Rad51 proteins but is independent of Rad18 and Rad5 ubiquitin ligases and PCNA ubiquitylation [164]. It seems that, at least in yeast, TS is the pathway of choice while TLS and HR appear to be backup mechanisms activated later in the cell cycle [164,165,166,167]. Moreover, while TS relies almost exclusively on the sister chromatid as a template for damage bypass, the HR machinery may use other DNA sequences as donors, potentially leading to the loss of heterozygosity or gross chromosomal rearrangements [168,169].

## 5. Mechanisms of Cohesin Recruitment to DNA Damage Sites

### 5.1. Targeting Yeast Cohesin to DNA Double-Strand Breaks

Induction of a single DSB in yeast cells arrested in the S or G2/M phase is sufficient to recruit cohesin specifically to a damage site. Cohesin enrichment around DSB sites depends on the Scc2-Scc4 complex, suggesting that cohesin accumulation in response to DNA damage is a result of de novo loading. Just 90 min after DSB induction, cohesin-rich domains can be found as far as ≈50 kb left and right from a damage site. Exceptions are the regions immediately adjacent to the DNA break, which probably undergo DNA resection and contain ssDNA [170,171]. How the cohesin loader and cohesin recognize DSB is not entirely clear, but it seems that optimal accumulation of cohesin at a damage site requires DDC proteins. It was shown that the MRX complex, together with Mec1 and Tel1 sensor kinases, is crucial for cohesin enrichment around the DSB. Moreover, cohesin association with chromatin around the DSB requires phosphorylation of histone H2A [170,172,173]. On the other hand, lack of Rad53 kinase activity mildly affects cohesin levels near the DSB, while there are conflicting data regarding Rad9 contribution to cohesin binding to chromatin in response to the DSB [172,173,174]. Since H2A phosphorylation is one of the earliest signals of DSB induction, that likely precedes cohesin accumulation, and because cohesin and H2A-P domains largely overlap, it is possible that Mec1- and Tel1-dependent histone H2A phosphorylation is the initial signal for cohesin recruitment to the DSB [89]. Finally, it was also shown that the RSC complex is important for cohesin binding to chromatin around the DSB; however, whether it resembles the role of RSC in DDC activation, generation of an optimal chromatin environment for the loading process, or both, is not yet known [44,175,176].

Under normal conditions cohesion establishment is linked to replication-coupled Smc3 acetylation by Eco1. Thus, cohesin loaded outside S-phase interacts with chromatin only transiently [59,60,61,63,64,65]. Remarkably, cohesin can become cohesive independently of DNA replication in the presence of DNA damage. It was found that cohesin complexes associated with chromatin exclusively after S-phase can stably hold sister chromatids together in the presence of a DSB. This damage-induced cohesion (DI cohesion) is generated not only around DSBs but genome-wide, including cohesin binding sites located on chromosomes where no DSBs are present [172,173]. Moreover, a cohesion assay performed on a strain lacking *RAD52* revealed that cohesion establishment after DSB generation is independent of DNA synthesis and replication/recombination intermediates associated with HR-dependent damage repair [172,173]. In contrast, DI cohesion induced in response to a single DSB was affected by disruption of Mre11, Mec1, and Chk1 [173]. Lack of *CHK1* resulted in normal cohesin binding patterns along chromosome arms and around DSBs, but prevented generation of genome-wide cohesion in response to DNA damage. It was found that in cells suffering DSBs, Chk1 phosphorylates the Scc1 cohesin subunit, most likely on Ser83, restored the ability to form cohesive cohesins. Interestingly, phosphomimic Ser83Asp mutation enabled cohesion establishment even without DSB induction, suggesting that Ser83 phosphorylation is sufficient for generation of replication-independent cohesion [177]. Further research revealed that Ser83 modification allows Eco1 to acetylate Scc1 probably on Lys84 and Lys210. This prevents Wpl1-dependent cohesin deposition and stabilizes cohesin on chromatin. Interestingly, Smc3 acetylation, which enables stable chromatid tethering in S-phase, does not allow cohesion to be established after DSB formation [178]. Furthermore, recent research revealed that Scc1 sumoylation is important for DI cohesion. Although SUMO-modified forms of Scc1 are also present under normal conditions and are required for cohesion establishment, their levels are greatly enhanced by DSB induction [179,180]. Scc1 sumoylation was shown to be catalyzed largely by the Mms21 SUMO ligase, a subunit of the cohesin-related Smc5/6 complex, and occurs independently of Chk1 or Eco1 activity. Lack of SUMO acceptor sites on Scc1 results in decreased levels of cohesin bound near DSB and impaired DI cohesion [180]. How cohesin sumoylation contributes to efficient cohesion is currently unknown. However, the multiplicity of cohesin posttranslational modifications underscores the complexity of cohesin regulation in response to different cellular contexts. Finally, recent research revealed that Pol η is required specifically for establishment of global cohesion but its role in genome-wide cohesion establishment is largely unknown [181].

Interestingly, postreplicative DSB induction not only promotes de novo cohesin loading around the break site and genome-wide cohesion establishment but also stimulates partial removal of the cohesin complexes in a separase-dependent manner. These include some of the cohesin complexes loaded in the recent S-phase but also those that became associated with chromatin in response to the DSB and were situated near the break site. The exact role of this process remains elusive, but it seems that cohesin presence in DSB-proximal regions may reduce chromatin accessibility for repair factors and disrupt damage repair [182].

### 5.2. Targeting Human Cohesin to DNA Double-Strand Breaks

As it was shown in budding yeast, human cells respond to a single DSB via the increased accumulation of cohesin near the break site [183,184,185]. However, in contrast to yeast, human cohesin domains are formed only proximally to DSB and do not extend beyond 5 kb from the breakpoint [183]. Cohesin enrichment around damage site depends on the NIPBL-MAU2 complex, suggesting that like in yeast cohesin accumulation in response to DNA damage is the result of de novo loading [184,186,187]. Recently, it has been demonstrated that the cohesin loading complex recruitment to DSB relies mostly on NIPBL interaction with HP1γ and requires the presence of RNF8-ubiquitin ligase [184]. Similar to what was revealed in yeast, accumulation of cohesin at damage sites depends on checkpoint proteins. It was shown by Kim et al. that the MRN complex is crucial for cohesin targeting to damaged regions and that MRN-dependent cohesin recruitment seems to be mediated by physical interaction between two complexes [188]. Later, it was shown that ATM kinase is crucial for increased cohesin levels in the presence of DNA damage. In response to ionizing radiation (IR), ATM phosphorylates SMC3 at Ser1083. This modification was shown to be required for increased genome-wide cohesin binding to chromatin after DNA damage [185,189]. Moreover, in response to IR, ATM and ATR activate ESCO1, enabling additional acetylation of SMC3 Lys105 and Lys106. As in the case of Ser1083 phosphorylation, SMC3 acetylation was shown to be important for global cohesin reinforcement after IR [185]. Like in yeast, human SCC1 can be sumoylated at multiple sites by MMS21; however, IR does not enhance SUMO-SCC1 levels. Moreover, in contrast to yeast, sumoylation of human SCC1 is not required for cohesin recruitment to damage sites or sister chromatid cohesion but rather has a function strictly in DNA repair [190]. 

### 5.3. Cohesin Recruitment and Dynamics at Stalled Replication Forks

In budding yeast, cohesin transiently accumulates at replication sites during normal S-phase. However, in response to replication inhibitors or replication blocking agents such as hydroxyurea (HU) or methyl methanesulfonate (MMS), respectively, additional cohesins bind to damage sites. This results in formation of the cohesin domain that extends up to ≈5 kb on each site of replication forks stalled at the early replication origin [47]. Importantly, increase of cohesin levels during replication stress was specific to active replication regions as no cohesin enrichment was found at late replication origins, centromeres, or between genes that are transcribed in converging directions. Cohesin enrichment at stalled replication forks depends on the cohesin loading complex and the MRX complex. However, neither Mec1 and Tel1 kinases nor H2A phosphorylation was required for cohesin binding to HU-arrested replication forks. In fact, only the *mec1*Δ *tel1*Δ double deletion led to a significant decrease of cohesin levels on chromatin [47,191]. This suggests that molecular requirements for cohesin accumulation at stalled replication forks may differ from the mechanisms determined for DSB-induced cohesin recruitment. It has been recently revealed that the Irc5 DNA translocase promotes cohesin binding to damage sites during replication stress. In the absence of *IRC5*, the cohesin complexes accumulate more slowly and to a lesser extent at early replication origins during MMS treatment. These data may point to an important role of chromatin structure in replication stress-induced cohesin enrichment at replication sites [191]. It was also shown that in response to HU treatment, Smc1, Smc3, and Scc1 cohesin subunits are ubiquitylated by the Rsp5^Bul2^ ubiquitin ligase (NEDD4 in humans) in a Mec1-dependent manner [192]. This induces cohesin interaction with the Cdc48 segregase (VCP or p97 in humans), that together with Wpl1, may open the cohesin ring and allow cohesin relocation behind the stalled replisome. These data suggest that cohesin complexes associated near stalled replication forks are in a dynamic state and likely require redistribution to enable replication resumption [192]. It is worth noting that little is known about the role of cohesin during replication stress in human cells. Nevertheless, induction of NIPBL and Smc1 phosphorylation or Scc1 ubiquitylation in response to UV, together with increased sensitivity of the Smc1^S957A,S966A^ mutant to MMS, suggest that also in humans cohesin may be required for DDR at blocked replication forks [193,194,195].

## 6. Role of Cohesin in DNA Damage Response

It has been shown that the cohesin loader and cohesin are crucial for survival in the presence of a wide range of genotoxins [190,191,193,194,196,197]. Moreover, while reduction of cohesin levels to 13% of wild type levels has no effect on sister chromatid cohesion, it leads to increased sensitivity to DNA damaging agents [198]. This result suggests that DDR pathways are especially sensitive to cohesin disruption, but the exact function of cohesin at DNA damage sites is still not entirely clear.

Genetic analysis revealed that during MMS-induced replication stress cohesin work in a common pathway with Rad18-mediated TS but independently of TLS or HR ([191] and data to be submitted for publication). In agreement with these data, it was shown that during MMS treatment, cohesin accumulates near stalled replication forks to promote efficient formation of X-shaped molecules that are crucial for TS. Consequently, lack of functional cohesin or cohesin regulators lead to decreased levels of sister chromatid junctions, resulting in accumulation of ssDNA gaps and the inability to complete DNA synthesis [47,72,191,192]. Recently, it has been demonstrated that artificial tethering of sister chromatids can suppress a TS defect in cohesin mutants and restores the generation of X-shaped molecules [72]. Together, these data indicate that the role of cohesin at stalled replication forks is to promote formation of sister chromatid junctions by holding chromatids in close proximity at the sites of perturbed replication. This allows timely and efficient resumption and completion of DNA replication, imposing the use of error-free TS instead of potentially mutagenic TLS or HR (Figure 3).

The exact role of cohesin in response to DSB induction is not known. In budding yeast, pulsed-field gel electrophoresis (PFGE) analysis revealed that lack of the cohesin loader or cohesin leads to decreased recovery of chromosomes after IR exposure [171,172,199]. This suggests that both complexes are required for efficient repair of DSBs. Nevertheless, which aspect of DSB repair is disrupted in cells lacking functional cohesin is still unknown. In budding yeast, it seems that in the absence of cohesin DNA resection occurs normally [170]. Remarkably, it is the prolonged cohesin occupancy near the break site that can interfere with the generation of 3′ overhangs, preventing timely repair of DNA damage [182]. In humans, in the absence of cohesin, RAD51 foci form normally in response to DSB suggesting that cohesin is not required for RAD51 accumulation on DNA [186]. On the other hand, recent studies revealed that in vitro SA1 binds to DNA structures that resemble replication forks or DNA overhangs, while PDS5B has a high affinity to the D-loop structure [200,201]. Moreover, PDS5B was shown to stimulate D-loop formation and annealing of complementary ssDNA. Furthermore, PDS5B interacts with core HR factors BRCA2, PALB2, and RAD51 [197,201]. These data suggest that PDS5B, and by extension cohesin, may facilitate the synaptic step of HR in humans (Figure 3).

Why yeast cells establish global cohesion in response to DSB is still unclear. It was proposed that induction of genome-wide cohesion may prevent premature sister chromatid separation [173]. However, while mutations in *ECO1* or lack of *POLη* results in increased incidence of chromosome loss, deletion of *CHK1* does not [173,181,202]. Moreover, *chk1*Δ or *rad30*Δ (*POLη-*deficient cells) cells are no more sensitive to IR than wild type cells also suggesting that global cohesion is not important for DNA repair [181,203,204]. These data may suggest the existence of parallel pathways that induce global cohesion and/or may reflect different mechanisms of genome-wide cohesion induction in response to a single DSB or IR.

Several studies in both yeast and humans suggest that cohesin regulates the choice of donor sequence for DNA repair. It was demonstrated that cohesin is not important for intrachromosomal recombination [170,205]. Instead, cohesin promotes the use of the sister chromatid as a template for recombination-based damage repair (Figure 3) [186,190,205,206]. Consequently, cohesin disruption results in increased recombination between homologous chromosomes [47,206]. How mechanistically cohesin imposes recombination between sister chromatids is not clear. In response to DSBs, mobility of damaged DNA increases, allowing broken DNA to explore large volumes of the nucleus to find a homologous donor. However, enhanced movement of chromatin stimulates recombination and may increase the likelihood of the use of ectopic sequences [207,208,209]. Dion et al. showed that in budding yeast, cohesin limits the movement of DNA repair foci under normal conditions and after DSB induction [209]. These data suggest that by constraining DNA mobility, cohesin may direct DNA repair to the sister chromatid and prevent the use of other donors. Interestingly, in human cells, cohesin blocks NHEJ and A-EJ of distant DSB ends, but not those situated close to each other [210]. This prevents chromosome fusions and generation of mutations that arise during end joining reactions. On the other hand, cohesin depletion leads to severe genome instability manifested in higher incidence of interchromosomal fusions as well as deletions, insertions, and duplications (Figure 3). The exact mechanism by which cohesin precludes chromosome fusions is not known, but rather than the general inhibition of NHEJ, cohesins may stabilize broken DNA ends to limit its mobility and decrease the likelihood of finding another free distal DNA end [210].

Finally, human cohesin plays a role in DNA damage checkpoint activation. In response to IR, ATM phosphorylates SMC1 at Ser957 and Ser966, and SMC3 at Ser1083. Both modifications require the presence of the intact NBS1 subunit of the MRN complex, and possibly BRCA1 in the case of SMC1 phosphorylation. It was shown that the inability to phosphorylate SMC1 or SMC3 leads to an S-phase checkpoint defect indicated by lack of DNA synthesis inhibition in the presence of DNA damage [189,194,211]. Moreover, later research revealed that cohesin is also required for G2/M DNA damage checkpoint activation in response to IR (Figure 3) [212]. It is not entirely clear how cohesin performs its checkpoint function. At least in the case of SMC1, mutation of phosphorylation sites does not influence cohesin accumulation at damage sites [211]. Interestingly, it seems that the cohesin complexes engaged in DDC do not have to be in a cohesive state. It has been suggested that cohesin may be required for efficient targeting of the 53BP1 adaptor to damage sites and full activation of CHK2 [212].

## 7. Summary

The cohesin complex is a key factor that mediates genomic stability since disruption of cohesin or cohesin regulators has been shown to lead to developmental disorders and cancers [213,214]. During a normal interphase, cohesin promotes equal segregation of genetic material preventing precocious separation of sister chromatids. In addition, cohesin regulates global transcription and influences chromosome condensation [10,215,216]. In recent years, an accumulating body of evidence demonstrates that also under genotoxic stress conditions, cohesin is crucial for the preservation of genomic stability. Cohesin facilitates DNA repair and favors the use of the sister chromatid during recombination-based processes, reducing the probability of heterozygosity loss or gross chromosomal rearrangements. During S-phase, cohesin limits the synthesis of damaged DNA and blocks the joining of distant broken DNA ends, preventing chromosome translocations. In agreement with those experimental data, it was shown that cell lines derived from patients with mutations in *NIPLB*, *SMC1*, or *SMC3* show increased sensitivity to DNA damaging agents [217,218,219,220].

Although much progress has been made in understanding the role of cohesin in the DNA damage response, there are some questions that need to be answered. First, how exactly does cohesin promote the repair of DNA damage? Is providing close proximity of sister chromatids the only role of cohesin in DNA damage repair, or do cohesin’s functions exceed simple embracing of DNA? Does the cohesin complex actively participate in DNA repair by HR? Finally, could cohesin target other repair proteins to damage sites regulating spatially and temporally their occurrence at DNA damage sites? Answers to these questions will allow us to better understand the crucial role of cohesin in the preservation of genome stability.

## Figures and Tables

**Figure 1 genes-09-00581-f001:**
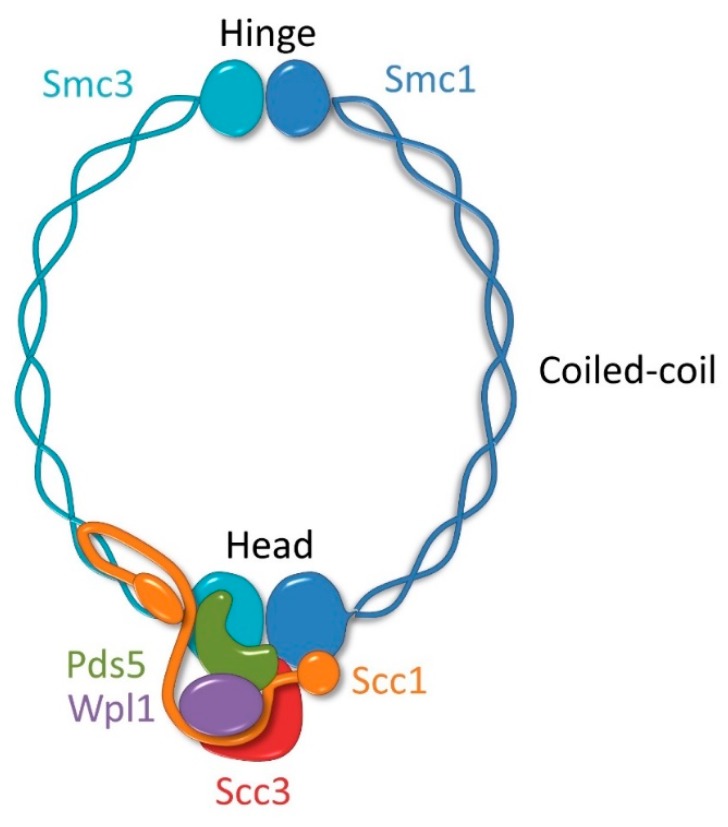
The cohesin complex. Cohesin is composed of Smc1 and Smc3 proteins that contain two globular domains, called the hinge and the head, separated by a long coiled-coil domain. Smc1 and Smc3 interact with each other through the hinge and the head domains. The hinge domain is the possible entry gate for the DNA while the head is an ATPase. Scc1 binds Smc3 and Smc1 with its N- and C-terminus, respectively, completing the cohesin ring. Pds5 and Scc3 are stably bound cohesin subunits that interact with cohesin through Scc1. Wpl1 binds to cohesin only temporarily through Scc3, Pds5, Scc1, and Smc3.

**Figure 2 genes-09-00581-f002:**
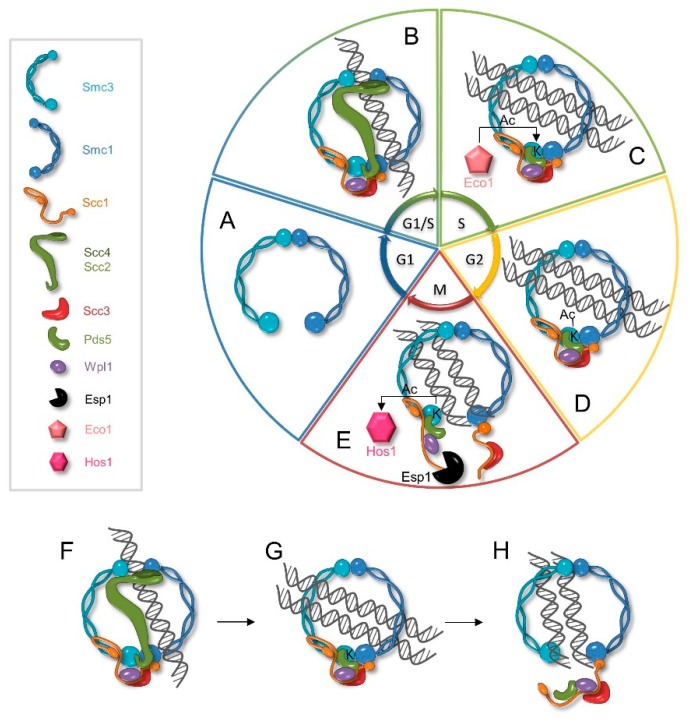
Possible cohesin cycle in yeast. In the G1 phase of the cell cycle, the cohesin complex does not form because Scc1 is absent (**A**). In the late G1/early S-phase Scc1 expression is reinduced allowing formation of the complete cohesin complex. Next, the chromatin-bound Scc2–Scc4 complex directs cohesin to specific chromosomal loci and facilitates cohesin loading reaction by creating an entry gate for the DNA, possibly by separating Smc1 and Smc3 hinge domains (**B**). During the S-phase, Eco1 acetylates some of the cohesins. This prevents Wpl1-dependent cohesin deposition and enables stable entrapment of sister chromatids (**C**), until mitosis cohesion is maintained by Pds5 and Scc3 (**D**). At the anaphase onset, the Esp1 separase cleaves Scc1, allowing Hos1 to deacetylate Smc3, which leads to cohesin removal from chromatin (**E**). Many cohesins loaded in G1/S or later (**F**) are not acetylated by Eco1 (**G**), and interact with the chromatin only transiently because Wpl1 imposes Scc1 dissociation from Smc3, creating an exit gate for DNA (**H**).

**Figure 3 genes-09-00581-f003:**
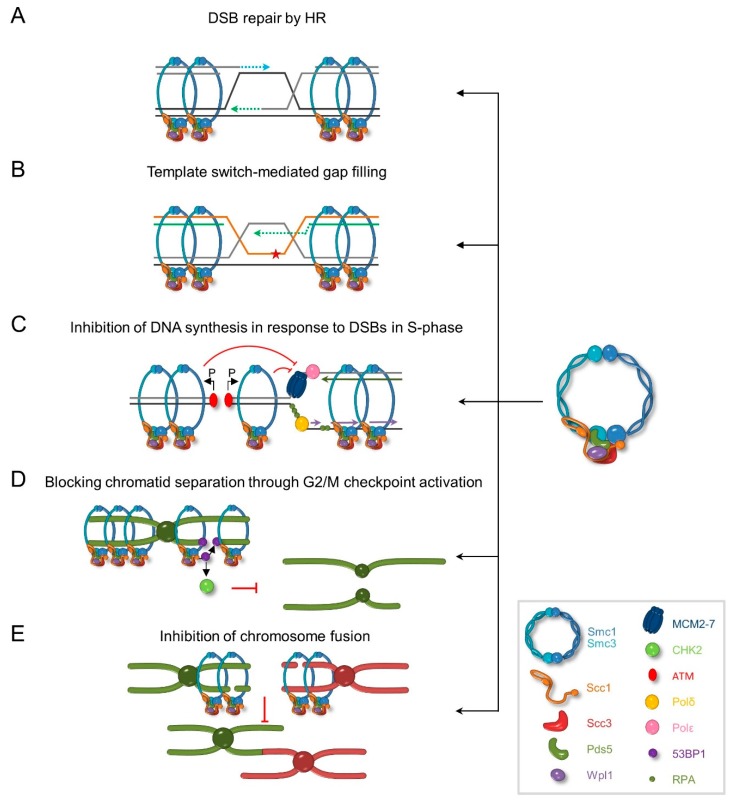
Cohesin functions in DNA damage response. (**A**) DSB induction leads to cohesin accumulation near the break site promoting efficient repair by homologous recombination (HR) using the sister chromatid as a template. (**B**) Cohesin enables replication completion by keeping sister chromatids in close proximity to allow template switch (TS). (**C**) In the presence of DNA double-strand breaks (DSB), cohesin becomes phosphorylated by ATM kinase and inhibits DNA synthesis. (**D**) Cohesin blocks premature entry to mitosis under DNA damage conditions allowing complete accumulation of 53PB adaptor and full activation of CHK2 kinase. (**E**) Cohesin prevents joining of distal DSBs.

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
