# Peer review of "The Emerging Role of Cohesin in the DNA Damage Response"

_genes, 2018, doi:10.3390/genes9120581_

Round 1

Reviewer 1 Report

Litwin et al reviews the role of cohesin in the DNA damage response by first introducing the composition and function of the cohesin complex in a clear manner. This is followed by an introduction to DNA damage responses. This is a very broad subject and thus it is described in much less details. The last parts of the manuscript describe the involvement of cohesin in the DNA damage response. 

The manuscript is well structured and well referenced and it covers an interesting area. However, there is a problem with numbering of references that starts around refs. 30-40. The draft contains three figures. Two of these are adequate but parts of fig. 3 does not clearly convey the message that was intended. Fig. 3 C and D is supposed to illustrate the involvement of cohesion in checkpoint activation but nothing on the illustration shows anything about checkpoint activation.  Moreover, it is not indicated what the pin, blue, yellow and green symbols in fig. 3C represent. The rview is based mainly on findings in yeast but also brings in knowledge from humans. Most places it is clear, which species is referred to but a few places this info is missing.

Minor comments.

P1. Line 7-8: The first sentence in the abstract should be rewritten since “any change in genetic information may result in genomic instability” is inaccurate

P2. Line 54: “WAP1 in humans” should be “WAPL in humans”. This should be implemented throughout the manuscript.

P2. Line 57 Figure 1: The color of Scc1text seems more yellow than the Scc1 protein that seems more orange. The colors should be more similar to prevent confusion.  

P3. Line 94: “hydrolase” should be “hydrolyse”

P5. Line 158: It should be specified more clearly that it is only sororin that is only present in vertebrates

P5. Line 159: WAPI should be WAPL

P5. Line 184-186. There should be another introduction sentence to this paragraph. The current sentence is somewhat detailed but still miss some important information and furthermore the HR process is described again later in the paragraph in a much better way.

P5. Line 187: “plays a minor in this process” should be “plays a minor role in this process”

P.6 Line 233-244: Here it is not clear that it is yeast proteins that are described.

P. 8 line 295-297: “Because accommodation of damaged DNA bases requires a more spacious active site, translesion polymerases lack proofreading activity. “
This sentence is inaccurate. Proof reading activity refers to the exonuclease activity, while the spacious active site gives a lower fidelity for base incorporation. 

P. 8 line 305: “TS. Template switch”… Throughout the manuscript it is not clear why the authors do not consistently use introduced abbreviation.

P. 8 line 325: “DSB” should be “DSBs”

P. 9 line 361: It should be clarified was is ment by “postreplicative cohesion block“

P. 9 line 383: “DSB” should be “DSBs”

P. 9 line 385: “It should be however noted” should be changed 

P. 9 line 390: “DSB” should be “DSBs”

P. 10 line 411: It is unclear what “unrelated converging intergenes” refers to

P. 12. 464: “In humans, in the absence of cohesin RAD51 foci form normally in response to DSB, suggesting that cohesin is not involved in the synaptic stage of HR“ This may be true but the authors should include a reference that shows that RAD51 foci can not represent presynaptic RAD51 nucleofilaments or otherwise rephrase

P. 12 line 465: I suggest to insert “On the other hand” before “Recent ..” to stress that the findings point in opposite directions

P. 12 line 485: “DSB” should be “DSBs”

P. 13. Line 517: “DNA damage repair” should be “DNA repair”

P. 13 Line 523-524: Throughout the manuscript it is not clear why the authors do not consistently use introduced abbreviation. DNA damage response or DDR

Author Response

Response to Referee #1 comments:

Litwin et al reviews the role of cohesin in the DNA damage response by first introducing the composition and function of the cohesin complex in a clear manner. This is followed by an introduction to DNA damage responses. This is a very broad subject and thus it is described in much less details. The last parts of the manuscript describe the involvement of cohesin in the DNA damage response.

The manuscript is well structured and well referenced and it covers an interesting area. However, there is a problem with numbering of references that starts around refs. 30-40.

We have carefully edited the numbering of the references and we corrected all misnumbered citations.

The draft contains three figures. Two of these are adequate but parts of fig. 3 does not clearly convey the message that was intended. Fig. 3 C and D is supposed to illustrate the involvement of cohesion in checkpoint activation but nothing on the illustration shows anything about checkpoint activation.  Moreover, it is not indicated what the pin, blue, yellow and green symbols in fig. 3C represent.

As suggested by the Referee we modified Figure 3C and 3D to better illustrate the involvement of cohesin in checkpoint activation. We also included a legend to describe what respective symbols represent.  

The rview is based mainly on findings in yeast but also brings in knowledge from humans. Most places it is clear, which species is referred to but a few places this info is missing.

This was corrected.

Minor comments.

P1. Line 7-8: The first sentence in the abstract should be rewritten since “any change in genetic information may result in genomic instability” is inaccurate

We corrected this as follows (p1, line 7-9):” Faithfull transmission of genetic material is crucial for all organisms since changes in genetic information may result in genomic instability that causes developmental disorders and cancers”.

P2. Line 54: “WAP1 in humans” should be “WAPL in humans”. This should be implemented throughout the manuscript.

This was corrected.

P2. Line 57 Figure 1: The color of Scc1 text seems more yellow than the Scc1 protein that seems more orange. The colors should be more similar to prevent confusion. 

We checked that the color used for Scc1 symbol is the same as the color used for “Scc1” caption.

P3. Line 94: “hydrolase” should be “hydrolyse”

This was corrected.

P5. Line 158: It should be specified more clearly that it is only sororin that is only present in vertebrates

This was corrected as follows (p5, line 167-172): "Later in the cell cycle, in yeast cohesion is sustained by Pds5 and Scc3. How these proteins maintain cohesion is not clear but Pds5 may protect Smc3 Lys112 and Lys113 from deacetylation and/or preclude polysumoylation of Scc1 and its proteosomal degradation (Figure 2). In humans, cohesion is maintained by Sororin, which is exclusively present in vertebrates. It was proposed that Sororin blocks WAPL binding to Pds5 preventing cohesin deposition from chromatin [69-72]."

P5. Line 159: WAPI should be WAPL

This was corrected.

P5. Line 184-186. There should be another introduction sentence to this paragraph. The current sentence is somewhat detailed but still miss some important information and furthermore the HR process is described again later in the paragraph in a much better way.

Taking into account suggestions of both Referees, we rewrote the whole section of “DNA damage response mechanisms”.

P5. Line 187: “plays a minor in this process” should be “plays a minor role in this process”

This was corrected.

P.6 Line 233-244: Here it is not clear that it is yeast proteins that are described.

We clearly assigned indicated proteins either to yeast or humans.

P. 8 line 295-297: “Becauseaccommodation of damaged DNA bases requires a more spacious active site, translesion polymeraseslack proofreading activity. “This sentence is inaccurate. Proof reading activity refers to the exonuclease activity, while the spacious active site gives a lower fidelity for base incorporation.

This was corrected as follows (p7, 287-291):” Next, monoubiquitylated PCNA attracts translesion polymerases that replicate across DNA lesion due to more specious active site that allows accommodation of variety of modified DNA bases. However, TLS polymerases lack proofreading activity, which leads to increased frequency of misincorporated nucleotides and largely accounts for genome-wide mutagenesis [145-148].

P. 8 line 305: “TS. Template switch”… Throughout the manuscript it is not clear why the authors do not consistently use introduced abbreviation.

This was corrected throughout the manuscript.

P. 8 line 325: “DSB” should be “DSBs”

This was corrected.

P. 9 line 361: It should be clarified was is ment by “postreplicative cohesion block“

We addressed this by rewriting the sentence as follows (p9, line 348-350): ”Interestingly, phosphomimic Ser83Asp mutation enabled cohesion establishment even without DSB induction, suggesting that Ser83 phosphorylation is sufficient for generation of replication-independent cohesion”.

P. 9 line 383: “DSB” should be “DSBs”

This was corrected.

P. 9 line 385: “It should be however noted” should be changed

It was corrected as follows (p9, line 374-376): “However, in contrast to yeast, human cohesin domains are formed only proximally to DSB and do not extend beyond 5 kb from the breakpoint [188]”.

P. 9 line 390: “DSB” should be “DSBs”

This was corrected.

P. 10 line 411: It is unclear what “unrelated converging intergenes” refers to

We clarify this as follows (p10, line 399-401):” Importantly, increase of cohesin levels during replication stress was specific to active replication regions as no cohesin enrichment was found at late replication origins, centromeres or between genes that are transcribed in converging directions”.

P. 12. 464: “In humans, in the absence of cohesin RAD51 foci form normally in response to DSB, suggesting that cohesin is not involved in the synaptic stage of HR“ This may be true but the authors should include a reference that shows that RAD51 foci can not represent presynaptic RAD51 nucleofilaments or otherwise rephrase

We agree with the Referee that Rad51 foci formation do not necessarily occurs in the synaptic stage of HR. To clarify this we rephrase this sentence as follows (p11, line 455-456): ”In humans, in the absence of cohesin RAD51 foci form normally in response to DSB, suggesting that cohesin is not important for RAD51 accumulation on the DNA”.

P. 12 line 465: I suggest to insert “On the other hand” before “Recent ..” to stress that the findings point in opposite directions

This was corrected.

P. 12 line 485: “DSB” should be “DSBs”

This was corrected.

P. 13. Line 517: “DNA damage repair” should be “DNA repair”

This was corrected.

P. 13 Line 523-524: Throughout the manuscript it is not clear why the authors do not consistently use introduced abbreviation. DNA damage response or DDR

This was corrected throughout the manuscript.

Reviewer 2 Report

”The emerging role of cohesin in the DNA damage response”, by Litwin et al, an invited review for a special issue in Genes, sets out to summarize the recent advances in understanding the function of cohesin in DNA damage signaling and repair.

The authors also indicate that they will give an overview of the architecture of the cohesin complex, and also of the molecular mechanisms for sister chromatid cohesion, as well as describing the main DNA damage response (DDR) pathways. All this together sounds extremely ambitious given the limited number of pages for the review (12). Overall the text in the manuscript is well written with good flow and the goals of the review are in essence reached, but I do have some suggestions and concerns.

To achieve a comprehensive review of the role of cohesin in DDR, backgrounds to both the mechanism of action for cohesin normally, both its architechture and its canonical functions, as well as briefly introduce the DDR are naturally needed. However, in its current form the manuscript teases the patience of a reader mainly interested in the subject suggested by the title. Actually 2 pages out of 12 describes and discusses the main focus of the paper. Instead it starts with a lengthy introduction to cohesin as well as to DNA damage response and repair, which is ambitious and very detailed, but never the less of rather basic nature, and very similar to multiple reviews about cohesin and DNA damage responses published during the last years. On the other hand, despite that the introduction is long and detailed it lacks any information or discussion about the recent findings regarding the involvement of cohesin in loop extrusion, of importance for the 3D formation of genomes. This could at least be mentioned and briefly discussed in relation to the function and actions of cohesin, if the aim is to describe the latest information about the architecture of cohesin, and the nature of its interactions with chromatin.

Furthermore, the knowledge about how the Scc1 and Smc3 subunits of cohesin interact based on structures of the same, could preferably be used to make Figure 1 more up to date.

The entire DNA damage response mechanism section is unnecessarily detailed, since the focus of the paper is the role of Cohesin in DDR, and how cohesin feeds into the details in the description of DSB repair and DNA damage tolerance mechanisms is not followed up upon  in the cohesin in DDR section. Therefore, this section could be shortened to allow greater focus on cohesin in relation to the different parts of the DNA damage response and repair.

A major problem with the entire manuscript after reference 35 is that the references are wrongly numbered. It is not the correct reference for the statement in the text. Could it be that one reference was added or deleted and that the rest are therefore mis-numbered in relation to the text. Or is the problem based on the fact that reference 35 has no title and journal details? these are instead given separately under number 36.

Another reference issue is ref 12, which is a review used as reference for a finding, that is of the same importance and magnitude as other findings described in this part of the text, where many original papers are cited. Why not give the original reference also in this case, at least also.

Why is the section about DNA damage checkpoint coming after the HR section? The checkpoint needs to be activated before the repair of DSBs.

Minor questions:

In the legend to figure 2B it is implicated that cohesin and Scc2-4 interact with each other before being loaded on chromatin together. This is an oversimplification, since as far as I have understood cohesin in many organisms, and also in budding yeast on arms, is very likely binding to preloaded Scc2-4.

Line 95: at the end there is an it to many

Line 109: NIPLB should be NIPBL

Line 193: Look at this sentence how can a nick be extended for kbs?

Line 320: add “the” before cell cycle and HR machinery

Line 507: Recent research … maybe not so recent, the correct reference nr 218 is from 2009.

Line 522: Enervald et al JEM 2013 could be added, extensive study on DNA damage sensitivity due to NIPBL and also SMC1 mutations

Author Response

Response to Referee #2 comments:

”The emerging role of cohesin in the DNA damage response”, by Litwin et al, an invited review for a special issue in Genes, sets out to summarize the recent advances in understanding the function of cohesin in DNA damage signaling and repair.

The authors also indicate that they will give an overview of the architecture of the cohesin complex, and also of the molecular mechanisms for sister chromatid cohesion, as well as describing the main DNA damage response (DDR) pathways. All this together sounds extremely ambitious given the limited number of pages for the review (12). Overall the text in the manuscript is well written with good flow and the goals of the review are in essence reached, but I do have some suggestions and concerns.

To achieve a comprehensive review of the role of cohesin in DDR, backgrounds to both the mechanism of action for cohesin normally, both its architechture and its canonical functions, as well as briefly introduce the DDR are naturally needed. However, in its current form the manuscript teases the patience of a reader mainly interested in the subject suggested by the title. Actually 2 pages out of 12 describes and discusses the main focus of the paper. Instead it starts with a lengthy introduction to cohesin as well as to DNA damage response and repair, which is ambitious and very detailed, but never the less of rather basic nature, and very similar to multiple reviews about cohesin and DNA damage responses published during the last years. On the other hand, despite that the introduction is long and detailed it lacks any information or discussion about the recent findings regarding the involvement of cohesin in loop extrusion, of importance for the 3D formation of genomes. This could at least be mentioned and briefly discussed in relation to the function and actions of cohesin, if the aim is to describe the latest information about the architecture of cohesin, and the nature of its interactions with chromatin.

According to Referees suggestions we mentioned about cohesin role in chromatin organization in Introduction section (p1, line 37-38). We also added a short paragraph (p4, line 142-148) describing cohesin ability to form DNA loops.

Furthermore, the knowledge about how the Scc1 and Smc3 subunits of cohesin interact based on structures of the same, could preferably be used to make Figure 1 more up to date.

According to Referees suggestions we modified Figure 1 updating the Scc1-Smc3 interface together with Pds5 and Scc3 schematic representation.

The entire DNA damage response mechanism section is unnecessarily detailed, since the focus of the paper is the role of Cohesin in DDR, and how cohesin feeds into the details in the description of DSB repair and DNA damage tolerance mechanisms is not followed up upon  in the cohesin in DDR section. Therefore, this section could be shortened to allow greater focus on cohesin in relation to the different parts of the DNA damage response and repair.

In agreement with Referee's suggestion we shortened the text removing unnecessary parts of “DNA damage response mechanism” section.

A major problem with the entire manuscript after reference 35 is that the references are wrongly numbered. It is not the correct reference for the statement in the text. Could it be that one reference was added or deleted and that the rest are therefore mis-numbered in relation to the text. Or is the problem based on the fact that reference 35 has no title and journal details? these are instead given separately under number 36.

We have carefully edited the numbering of the references and we corrected all misnumbered citations.

Another reference issue is ref 12, which is a review used as reference for a finding, that is of the same importance and magnitude as other findings described in this part of the text, where many original papers are cited. Why not give the original reference also in this case, at least also.

This was corrected.

Why is the section about DNA damage checkpoint coming after the HR section? The checkpoint needs to be activated before the repair of DSBs.

Acording to Referees suggestions we moved “DNA damage checkpoint activation in response to DSB” section before “Repair of DSBs by homologous recombination in mitotic cells” section.

Minor questions:

In the legend to figure 2B it is implicated that cohesin and Scc2-4 interact with each other before being loaded on chromatin together. This is an oversimplification, since as far as I have understood cohesin in many organisms, and also in budding yeast on arms, is very likely binding to preloaded Scc2-4.

We clarify this as follows (Fig 2B Legend): “Next, chromatin-bound Scc2-Scc4 complex directs cohesin to specific chromosomal loci and facilitates cohesin loading reaction by creating an entry gate for the DNA, possibly by separating Smc1 and Smc3 hinge domains”.

Line 95: at the end there is an it to many

This was corrected.

Line 109: NIPLB should be NIPBL

This was corrected.

Line 193: Look at this sentence how can a nick be extended for kbs?

This was corrected.

Line 320: add “the” before cell cycle and HR machinery

This was corrected.

Line 507: Recent research … maybe not so recent, the correct reference nr 218 is from 2009.

This was corrected.

Line 522: Enervald et al JEM 2013 could be added, extensive study on DNA damage sensitivity due to NIPBL and also SMC1 mutations

Enervald et al was added to the literature.